Isolation and identification of marine strains of Stenotrophomona maltophilia with high chitinolytic activity

Salas-Ovilla Roger
Gálvez-López Didiana
Vázquez-Ovando Alfredo
Salvador-Figueroa Miguel
Rosas-Quijano Raymundo raymundo.rosas@unach.mx rrquijano@yahoo.fr
Instituto de Biociencias, Universidad Autónoma de Chiapas , Tapachula , Chiapas , Mexico
Maurice Corinne
Electronic publication date: 2019 Jan 3
Publication date: 2019
Volume: 7
Electronic Location ID: e6102
Received 2018 Aug 27; Accepted 2018 Nov 6
Copyright: ©2019 Salas-Ovilla et al.
Copyright year: 2019
Copyright holder: Salas-Ovilla et al.
License: This is an open access article distributed under the terms of the Creative Commons Attribution License, which permits unrestricted use, distribution, reproduction and adaptation in any medium and for any purpose provided that it is properly attributed. For attribution, the original author(s), title, publication source (PeerJ) and either DOI or URL of the article must be cited.
License URL: https://creativecommons.org/licenses/by/4.0/

Keywords: Chitinolytic strains, Seawater, Chitinases, Specific activity, Shrimp residues

Funding: Mexican project SEP-PRODEP DSA/103.5/16/14474 CONACYT INFRA-2015-01-254256 This work was supported by the Mexican project SEP-PRODEP DSA/103.5/16/14474 with financing and the scholarship granted. Also, the project CONACYT INFRA-2015-01-254256 provided financing for the infrastructure used in this study. The funders had no role in study design, data collection and analysis, decision to publish, or preparation of the manuscript.

==============================
Chitin is the second most abundant organic compound in nature and represents a rich carbon and nitrogen source that is primarily transformed by bacterial communities. Bacteria capable of gradually hydrolyzing chitin into N-acetylglucosamine monomers can have applications in the transformation of residues from shrimp and other crustaceans. The objective of the present study was to isolate, characterize and identify microorganisms with high chitinolytic activity. These microorganisms were isolated and characterized based on macro- and microscopic morphological traits. Strains were selected on colloidal chitin agar medium primarily based on a hydrolysis halo larger than 2 mm and a growing phase no longer than 6 days. Secondary selection consisted of semi-quantitative evaluation of chitinolytic activity with a drop dilution assay. From the above, ten strains were selected. Then, strain-specific activity was evaluated. The B4 strain showed the highest specific activity, which was 6,677.07 U/mg protein. Molecular identification indicated that the isolated strains belong to the species Stenotrophomonas maltophilia.

Introduction

Shrimp production in Latin America was estimated to be between 500,000 and 600,000 tons in 2016, with Mexico being one of the main suppliers (FAO, 2016). Although the commercialization of this crustacean generated an economic revenue of more than 16 million pesos in 2016, the residues generated by the shrimp industry have a negative impact on the environment (SIAP, 2016). Of these residues, approximately 5% is transformed into products such as flours and extracts, which serve as a base for animal feed (Laxman et al., 2016). Shrimp rubbish containing 40% chitin, which is a polysaccharide composed of N-acetylglucosamine units (Younes & Rinaudo, 2015), represents an important primary resource for the production of bioactive molecules (Gao et al., 2016). Currently, there are two pathways for the production of chitin oligosaccharides: the chemical pathway and the biotechnological pathway. Although the most commonly used is the chemical pathway, it has negative implications: the cost of processing and the damage to the environment by highly corrosive chemical reagents (Kaur & Singh, 2013; Abirami et al., 2016). In contrast, the biotechnological pathway is an environmentally friendly process (Pal et al., 2014). In this context, the use of chitinases plays a key role (Wang, Song & Zhang, 2016). Chitinases are glycosyl-hydrolase proteins (EC 3.2.2.14) that cleave the β-1,4 bonds of the N-acetylglucosamine units, catalyzing chitin degradation (Deeba et al., 2016). Chitinolytic enzymes are synthesized in a wide variety of organisms, with fungi and bacteria being the most common. Although the chitinolytic activity of some species has been reported in the literature, it remains unknown which group of microorganisms is the most effective at decomposing this polymer. Several authors have reported marine ecosystems as a main source of chitinase-producing microorganisms, mainly bacteria (Sara et al., 2016; Swiontek et al., 2014; Suresh, 2012). Chitinolytic bacteria represent only 4% of the currently known bacteria (Swiontek et al., 2014). There are reports about chitin being degraded in aquatic environments by bacteria of the genera Aeromonas, Enterobacter, Chromobacterium, Arthrobacter, Flavobacterium, Serratia, Bacillus, Erwinia, and Vibrio (Souza et al., 2011). Other studies have isolated genera such as Eubacterium, Streptococcus and Clostridium from whale residues (Olsen et al., 2000); the species Bacillus licheniformis from food industry liquid residues (Laribi-Habchi et al., 2015); other genera such as Serratia and Streptomyces from crustacean residues (Castro et al., 2011); and the species Acinetobacter johnsonii and Bacillus amyloliquefaciens from shrimp residues (Imanda & Suharjono, 2015). However, the isolation of chitinolytic microorganisms in marine environments has been rare in comparison to the isolation of such microorganisms from terrestrial environments, due the most of reported chitinolitic microorganisms are aerobics (Swiontek et al., 2014). Therefore, the objective of the present study was to isolate and identify chitinolytic bacteria from shrimp residues and sea water.

Materials and Methods

Sample collection

Two kinds of samples were collected: the first consisted of 100 g of shrimp rubbish, and the second consisted of 6 samples of 200 ml of sea water from the sediment-free surface in different sectors of Las Escolleras-Puerto Madero in Chiapas, Mexico (latitude 14°42′19″N and longitude 92°24′28″W). The samples were aseptically collected in a hermetically sealed sterile cover and transported on ice container to the laboratory of the Biosciences Institute at Chiapas Autonomous University, where they were processed immediately.

Colloidal chitin preparation

Colloidal chitin was prepared according with Arnold & Solomon (1986) as follows: 10 g of commercial chitin from shrimp crust (Sigma-Aldrich, USA) was mixed with 100 ml of concentrated HCl and stored at 4 °C for 24 h. Then, 100 ml of distilled water was added, and the mixture was allowed to rest for 5 min. The chitin solution was then filtered with Whatman™ No. 1 paper, and the sediment was rinsed three times with distilled water. The chitin sediment was stored in a flask at 4 °C for later use.

Isolation of chitinolytic microorganisms

The samples were inoculated in serial dilutions by dispersion in the semisynthetic media composed by (g/l): Na2HPO4, 6.0; KH2PO4, 3.0; NH4Cl, 1.0; NaCl, 0.5; yeast extract, 0.05; agar, 15; colloidal chitin, 10 and pH 7 (Shivalee et al., 2016). All the reagents used were of analytical grade. Petri dishes were incubated at 30 °C for 4–5 days, and growth was checked daily. Microorganism colonies that grew were selected. Then, the chitinolytic activity was confirmed by the formation of a hydrolysis halo. The colonies’ purity was confirmed by reinoculation onto colloidal chitin agar medium.

Selection of chitinolytic bacteria

The primary isolation consisted of selecting the strains with the highest chitinolytic activity using the growing time of the colony and the hydrolysis halo size (2 mm at minimum) as parameters. For this purpose, the strains were inoculated onto colloidal chitin agar (3%) and incubated at 30 °C for 5 days. From this selection, a semi-quantitative assay was performed via a drop assay in colloidal chitin medium in the following manner: the primary isolates were inoculated in 2 ml Eppendorf tubes containing 1 ml of liquid colloidal chitin medium and incubated at 150 rpm at 30 °C for 3 days. Then, serial dilutions were made from each tube in saline solution (0.8% p/v), and the 10−6, 10−7 and 10−8 dilutions were selected and inoculated in duplicate.

Determination of specific chitinolytic activity

Chitinolytic activity was determined according to the method proposed by Chakrabortty, Bhattacharya & Das (2012). For this, all strains were cultured during five days in the semisynthetic broth media. After, supernatants from the respective cultures were obtained by centrifugation at 2,600 rcf for 5 min. The individual reaction mixture consisted of 1ml of individual culture supernatants, 1 ml of 1% (w/v) colloidal chitin in citrate phosphate buffer pH 5.5 and incubated at 50 °C for 30 min. Following incubation, all the reaction mixtures were put in boiling water bath for 3 min to stop the enzyme action. The solutions were centrifuged at 2,600 rcf for 10 min. The amount of reducing sugar in the supernatants (resulting due the chitinolytic activity) was determined by dinitrosalycilic acid (DNS) method. The absorbance was read at 540 nm using a UV–VIS spectrophotometer (SANYO Gallenkemp, Germany). To estimate the amount of protein, the Lowry method was used (Lowry et al., 1951). One enzymatic unit was defined as the amount of enzyme that produced 1 µmol of N-acetyl-D-glucosamine per minute.

Molecular identification of microorganisms

For identification, it was necessary to obtain the genomic DNA of the microorganisms. The strains were grown in 15 ml Falcon tubes with 5 ml of colloidal chitin medium and incubated at 150 rpm for 3 days at room temperature. Then, 1 ml of each cell culture was transferred to 1 ml sterile Eppendorf tubes, which were then centrifuged at 15,000 rcf for 5 min. The supernatant was discarded, and the cell pellet was rinsed three times with 1,000 µl of NaCl-EDTA (30 mM NaCl, 2 mM EDTA, pH 8.0) and then stored at 4 °C for later processing. The gDNA was extracted according to the method reported by Sachinandan et al. (2010) and was then quantified and stored. Then, a 16S ribosomal subunit gene fragment was amplified following the procedures described by Setia and Suhurjono (Imanda & Suharjono, 2015). The purified amplicons were sent to the Institute of Biotechnology of UNAM in Mexico for Sanger sequencing. The sequences were visualized and aligned in BioEdit v.7.0.5 software, and a phylogenetic tree was constructed in MEGA v.7.0.26 software. The evolutionary distance was calculated using 1,000 bootstrap replicates.

Statistical analysis

Analysis of variance (ANOVA) was performed on the specific chitinolytic activity data using the statistical software InfoStat v.11.7. Means were compared using Tukey tests (P ≤ 0.05).

Results

More than 200 colony-forming units (CFUs) from chitinolytic microorganisms were isolated (approximately 65% from solid residues and 35% from water), which presented mainly a yellow and white coloration. However, there were colonies with intense yellow and violet colorations, and some had dusty characteristics and the presence of mycelia. Then, the number of strains was reduced to 20 based on the colonial morphology criteria, and these colonies were purified. Through the selection procedure described in the Materials and Methods section, the number was reduced to 10 strains.

The strains B2, B3, B5, B8, B9 and B10 were isolated from shrimp shells, and the strains B1, B4, B6 and B7 were isolated from seawater. Most of the bacterial colonies presented similar macroscopic characteristics: circular shape, full borders, convex and smooth texture and white-yellow coloration. The 10 selected strains were Gram negative, bacillus shaped, individual and lightly curved.

The 10 strains expressed different hydrolysis halo sizes (Fig. 1). This is indicative of the isolated and selected organisms being different, even if they share morphological traits. Among the strains, the B4 strain showed the largest hydrolysis halo after 4 days of growth (0.9 mm).

Figure 1 Hydrolysis halos in colloidal chitin agar medium of the strains selected by drop dilution assay.

The brown squares show the 10 selected strains, whereas the white squares show the nonselected strains.

Specific chitinolytic activity

The results from the analysis of chitinolytic activity are shown in Fig. 2. These results indicated that the 10 selected strains have different capacities to hydrolyze chitin. The significant differences (P < 0.05) in the chitinolytic activity of the strains allowed the strains to be organized into three groups (Fig. 2). The first group was formed by the strains B2, B3, B8, B9 and B10, which presented the lowest specific activity; the second group was formed by the strains B1 and B6, which presented moderate activity; and the third group was formed by the strains B4, B5 and B7, which presented the highest specific activity.

Figure 2 Specific chitinolytic activity of the selected strains.

Values with the same letters (A, B or C) are significantly similar (P > 0.05).

The results of the specific activity test showed that the specific activity of the selected strains ranged from 1,593.87 U/mg to 6,677.07 U/mg of protein. In general, it was observed that there was a direct correlation between the observed specific activity and the formation of the hydrolysis halo in the plaque observed for each strain.

Amplification and sequencing of the 16S ribosomal subunit gene

When compared to sequences in the database, the sequences of the amplicons from strains B1 through B9 showed 99% identity with Stenotrophomonas maltophilia. The exact B10 strain sequence was unclear according to the analysis of the electropherogram, most likely because B10 was not a pure strain. To strengthen the identification of the strains, the phylogenetic relationship of the 16S ribosomal subunit gene of the strains in this study with the genera reported in the NCBI database was determined. The phylogenetic tree shown in Fig. 3 confirmed the identification of the bacterial strains as Stenotrophomonas maltophilia. Upon analyzing the strains in the database, it was observed that among the same species, there was genetic diversity, with 4 distinct groups. Most of the strains from this study are classified in group I, and the others are in groups II and III; none are in the largest group, group IV. On the other hand, the external reference strains (K-12, KT2440 and NC7401) were grouped in a distant branch of the genus Stenotrophomonas. This confirms that the strains isolated in this study are S. maltophilia.

Figure 3 Phylogenetic tree of the selected strains B1 to B9 and the reference strains constructed using the Maximum Likelihood algorithm.

Phylogenetic tree of the selected strains B1 to B9 and the reference strains constructed using the Maximum Likelihood algorithm.

Discussion

This research reports the isolation and molecular identification of 10 strains, all belonging to the genus Stenotrophomonas maltophilia with unusual chitinolytic activity, from a marine environment, shortly explored. The number of isolates obtained in this study was similar to that reported by Lilja (2013), who isolated more than 300 strains with chitinolytic activity from marine sources. Additionally, that author concluded that the types of samples, the treatment of the samples and the nutrimental composition of the culture media are critical factors that strongly influence the number of organisms obtained. This is due to the distribution of bacterial populations not being homogeneous in the samples and each organism having different nutritional requirements. Rashad et al. (2015) reported that shrimp samples tend to have a larger number of bacterial colonies (60% of the isolates), which aligns with what was found in this study. Even the percentage was maintained in the organisms with higher chitinolytic capacity. Chitin as a carbon and nitrogen source requires transformation by enzymes called chitinases into shorter oligomers that can be absorbed by microorganisms. The appearance of a halo surrounding the strains is indicative of chitinase activity. Previous studies have indicated that halo appearance requires a long incubation time between 5 and 6 days (Cody, 1989). The presence and isolation of Gram-negative bacteria is common in marine samples, especially shrimp residues. Gram-negative bacteria are the cause of diseases in crustaceans, which mainly attack the cuticle of these organisms (Sharmila et al., 2014). Vincy et al. (2014) isolated marine bacteria that formed dull, creamy yellow colonies and were Gram-negative and bacillus-shaped. On the other hand, Das, Lyla & Khan (2006) indicated that Gram-negative microorganisms have certain traits, such as tolerance to extreme temperature and rapid adaptation to different salt concentrations and nutrient deficiencies that allow them to survive in the marine environment. Therefore, the diversity of Gram-negative bacteria in marine environments is extremely high (90%) compared to that of Gram-positive bacteria (Nocker, Lepo & Snyder, 2004). This is the possible reason why all the selected bacteria where Gram negative.

With respect to the specific enzymatic activity found, the 10 strains showed values higher than those reported by Kim-Chi et al. (2011), who determined a maximum specific enzymatic activity of 18.33 U/mg of protein in a strain of Streptomyces sp. On the other hand, Swiontek & Donderski (2006) obtained specific chitinolytic activity values of 0.12 U/mg of protein, which is lower than the values obtained in this study. In Bacillus sp. bacteria, the strain Hu1 showed in crude extract an activity of 11.1 U/mg of protein (Dai et al., 2011), whereas another chitinolytic enzyme from B. licheniformis strain LHH100 showed a value of 494.5 U/mg of protein (Laribi-Habchi et al., 2015). Finally, Amar et al. (2017) studied the specific activity of a chitinase in Bacillus sp. strain R2 and found 234.1 U/mg of protein as the highest value for that strain. These findings indicate that the strains isolated and selected in this study have a strong chitinolytic capacity. However, the chitinolytic activity found in the present study is lower than that of the enzymes found by Liang, Hsieh & Wang (2013) in B. cereus with specific activities of 16,598 U/mg of protein.

The taxonomic classification of S. maltophilia was described recently when Palleroni & Bradbury (1993) proposed the creation of a new genus denominated Stenotrophomonas (“Stenos”, from the Greek word for narrow; “trophos”, from the Greek word for food; and “monas”, from the Greek word for unique or unity; thus, the name translates into “low substrates for food”). In the species name of S. maltophilia, “malt” is derived from the English word “maltose”, and “philia” is derived from the Greek word for friendship, that includes 10 species. These species are colonizers, which enables their isolation from diverse sources, such as water, sediment, soil, the rhizosphere, and plant tissues. However, there is scarce information about marine samples, and some authors report that S. maltophilia does not withstand extreme salinity levels, suggesting that it is not possible to isolate it from marine environments (Kielak et al., 2013; Zhu et al., 2011). Such reports are very relevant to this study, since the frequency of this microorganism in the isolates was almost 100%. This high frequency can be attributed to the microorganism’s high capacity to break down chitin as its carbon source and to its ability to displace other species during selection. Another factor to consider is that the pollution in the places from which the samples were taken could have influenced the presence of this microorganism. Contamination from the extensive release of organic residues by the industry, the accumulation of decomposing organisms, the presence of hydrocarbons from the use of fuels, and the presence of agrochemicals create favorable conditions for the growth of S. maltophilia. According to some reports, S. maltophilia is frequently present in environments polluted with hydrocarbons, pesticides and heavy metals, and its isolation is possible from these locations (Alfonso et al., 2016; Mukherjee & Roy, 2016; Ozdal et al., 2017). The identification, sequencing or use of the genetic material of these microorganisms could represent a biotechnological alternative to develop nonpathogenic microorganisms capable of degrading chitin.

Conclusion

More than 200 CFUs from bacteria with chitinolytic potential were isolated, from which 10 strains presented the desired traits regarding chitin degradation. Through molecular analysis of the 10 selected strains, it was determined that 9 belong to the species Stenotrophomonas maltophilia. The strain with the highest chitinolytic capacity was B4, with a specific activity of 6,677 U/mg of protein. The strains from Stenotrophomonas maltophilia identified in this study are excellent producers of chitinases, and the sequences chitin genes could be identified, sequenced, or used by heterologous gene expression purposes.

Supplemental Information

Supplemental Information 1 Specific Chitinolytic activity

Raw data of chitinolytic activity measured.

Click here for additional data file.

Additional Information and Declarations

Competing Interests

Author Contributions

Data Availability

The authors declare there are no competing interests.

Roger Salas-Ovilla conceived and designed the experiments, performed the experiments, analyzed the data, prepared figures and/or tables, approved the final draft.

Didiana Gálvez-López performed the experiments, analyzed the data, contributed reagents/materials/analysis tools, prepared figures and/or tables, authored or reviewed drafts of the paper, approved the final draft.

Alfredo Vázquez-Ovando analyzed the data, prepared figures and/or tables, authored or reviewed drafts of the paper, approved the final draft.

Miguel Salvador-Figueroa analyzed the data, contributed reagents/materials/analysis tools, authored or reviewed drafts of the paper, approved the final draft.

Raymundo Rosas-Quijano conceived and designed the experiments, performed the experiments, analyzed the data, contributed reagents/materials/analysis tools, prepared figures and/or tables, authored or reviewed drafts of the paper, approved the final draft, this author got the financial assistance.

The following information was supplied regarding data availability: NCBI: https://www.ncbi.nlm.nih.gov/nuccore/?term=Stenotrophomonas+maltophilia+rosas-quijano.

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
