# Peer review of "Isolation and identification of marine strains of Stenotrophomona maltophilia with high chitinolytic activity"

_PeerJ, doi:10.7717/peerj.6102_

## Round 0.1 · original submission · Minor Revisions

I believe the changes suggested by the reviewers are straightforward, but don't hesitate to reach out should you require any clarification.
Best,

Reviewer 1 ·

Basic reporting

no comment'

Experimental design

no comment

Validity of the findings

Some improvement in the comparison of enzyme activity data is required. No new experiment suggestion but authors should compare their enzyme activity data more appropriately.

Additional comments

Manuscript by Roger Salas-Ovilla et al is an interesting study where authors have isolated microorganism with high chitinolytic activity from the marine source. Authors have characterized the isolated strains by analyzing morphological and molecular properties. Mostly isolated strains belong to Stenotrophomonas maltophilia. Authors have given sufficient literature background. Articles included appropriate figures, tables, and raw data. The article is easy to follow.

I have the following comments.
1) Comparison of enzyme activity data:
(a) The authors have defined the unit as “One enzymatic unit was defined as the amount of enzyme that produced 1 μmol of N-acetyl-D-97 glucosamine per minute under standard conditions.”
(b) Liang et al definition: One unit of enzyme activity was defined as the amount of enzyme that produced 1 μmol of N-acetylglucosamine per min
(c) Kim-Chi Hoang et al definition: One unit of chitinase activity was defined as capable of releasing reducing ends corresponding to 1 μg of N-acetyl-D-glucosamine from glycol chitosan in an hour.
(d) Swiontek: The mentioned unit is not found in the referred paper.
As enzyme activity definitions are not uniform across the referenced papers, authors should reanalyze the data to make their points clear.

2) As authors have mentioned that the maximum reported enzymatic unit of 16,598 U/mg of protein, so in what sense isolated strains are more important? This explanation is required as authors have proposed using isolated strains for the biotechnological purposes. In what sense these isolated strains are more meaningful for the industry? Is the stain with the maximum reported unit not available commercially or is it not shared by that author? What will authors do to make their strains publicly available? Alternatively, authors should remove the proposal of its use in the industry if it fails on that ground.

3) Excel data: Authors have reported the specific activity in values with five digits after the decimal. Such accuracy is not possible. Authors should round off the values to the nearest whole number. Word media is a typo error. Values are average of three readings and should be written as “average”. It is not even the median.

4) The number of strains finally analyzed by authors should be mentioned in the abstract.

Minor comments
1) 1,000 ul of NaCl-EDTA: composition required for this
2) at 10,000 rpm for 5 min.: Authors should report the value in RCF instead of RPM. Since different centrifuge machines may have different rotor sizes, so RCF is more reliable value.
3) Swiontek BM, Donderski W. 2006. Chitinolytic bacteria in two lakes of different trophic status. Polish Journal of Ecology. 54(2):295-301. DOI: 381 https://doi.org/10.1016/j.soilbio.2017.09.019. Doi link is not leading to the right article.

·

Basic reporting

no comment

Experimental design

no comment

Validity of the findings

no comment

Additional comments

In the current manuscript authors have successfully isolated and identified chitinolytic bacteria
from shrimp residues and sea water. During first round of screening, more than 200 colony-forming units from chitinolytic microorganisms were isolated, after subsequent screening steps 10 colonies with high chitinase activities were selected for further characterization. Colonies on plate B4 demonstrated highest activities. 16S sequencing revealed that these bacterial colonies belong to belong to the species Stenotrophomonas maltophilia. Overall, manuscript is beautifully written, experiments are properly executed and results are well interpreted. I recommend this manuscript for acceptance in its current form.

Reviewer 3 ·

Basic reporting

The article is clear and literature references are sufficient for the background.

Experimental design

no comment

Validity of the findings

No comment

Additional comments

The aim of this paper was to isolate and identification of marine strains with chitinolytic activity. However, the scientific merit of this paper would be enhanced if the authors can address the following comments on the manuscript.

Introduction
1. Line 55-57: Author mentioned isolation of chitinolytic microorganisms in marine environments has been rare in comparison to the terrestrial environment. Is there any specific reason? Explain it in a few sentences.

Materials and Methods
1. Line 61-67: Author collected shrimp rubbish and seawater. Did they process the samples immediately or stored? If yes which condition and temperature before processing?
2. Line 69-74: Author described the colloidal chitin preparation. Did they follow specific protocol? If yes mention reference/s.
3. Line 76-81: Experiment used semisynthetic media to grow the microorganisms. However, readers can understand more if the author includes detail information about media.
4. Line 93-97: Chitinolytic activity was determined. Please explain this method in details.

Results
1. Line 134-145: Chitinolytic activity was determined and separated into three groups; lowest, moderate and highest specific activity groups. How were three groups differentiated?


Discussion
1. At the beginning of the discussion session, if the authors can start this section with their major findings, it would be more impactful.

Conclusion
Adding up the take-home message as well as the rationale of the findings or future directions would enhance the contribution of this study to the scientific community.

---

## Round 0.2 · accepted · Accept

All the points raised by the reviewers were adequately addressed. Please make sure to adequately read your proof once you receive it, as there are a few remaining typos, especially in the revised portions of the manuscript.